# Fusing Multiview Functional Brain Networks by Joint Embedding for Brain Disease Identification

**DOI:** 10.3390/jpm13020251

**Published:** 2023-01-29

**Authors:** Chengcheng Wang, Limei Zhang, Jinshan Zhang, Lishan Qiao, Mingxia Liu

**Affiliations:** 1School of Mathematics Science, Liaocheng University, Liaocheng 252000, China; 2School of Computer Science and Technology, Shandong Jianzhu University, Jinan 250101, China; 3College of Mathematics and Statistics, Sichuan University of Science and Engineering, Zigong 643000, China; 4Department of Radiology and BRIC, University of North Carolina at Chapel Hill, Chapel Hill, NC 27599, USA

**Keywords:** functional brain network, fusion, tensor factorization, autism spectrum disorder

## Abstract

**Background**: Functional brain networks (FBNs) derived from resting-state functional MRI (rs-fMRI) have shown great potential in identifying brain disorders, such as autistic spectrum disorder (ASD). Therefore, many FBN estimation methods have been proposed in recent years. Most existing methods only model the functional connections between brain regions of interest (ROIs) from a single view (e.g., by estimating FBNs through a specific strategy), failing to capture the complex interactions among ROIs in the brain. **Methods**: To address this problem, we propose fusion of multiview FBNs through joint embedding, which can make full use of the common information of multiview FBNs estimated by different strategies. More specifically, we first stack the adjacency matrices of FBNs estimated by different methods into a tensor and use tensor factorization to learn the joint embedding (i.e., a common factor of all FBNs) for each ROI. Then, we use Pearson’s correlation to calculate the connections between each embedded ROI in order to reconstruct a new FBN. **Results**: Experimental results obtained on the public ABIDE dataset with rs-fMRI data reveal that our method is superior to several state-of-the-art methods in automated ASD diagnosis. Moreover, by exploring FBN “features” that contributed most to ASD identification, we discovered potential biomarkers for ASD diagnosis. The proposed framework achieves an accuracy of 74.46%, which is generally better than the compared individual FBN methods. In addition, our method achieves the best performance compared to other multinetwork methods, i.e., an accuracy improvement of at least 2.72%. **Conclusions**: We present a multiview FBN fusion strategy through joint embedding for fMRI-based ASD identification. The proposed fusion method has an elegant theoretical explanation from the perspective of eigenvector centrality.

## 1. Introduction

Autism spectrum disorder (ASD) refers to a range of neurodevelopmental conditions characterized by social impairment, language difficulty, abnormal behavior, etc. [1,2,3,4]. Without timely and effective treatment, children with ASD tend to suffer from lifelong physical and mental health problems [5,6], resulting in a considerable burden on their families and society. Previous research has shown that early intervention can lead to positive outcomes for people with ASD later in life [7,8,9,10]. Therefore, accurate early detection of ASD is crucial in clinical practice.

Resting-state functional magnetic resonance imaging (rs-fMRI), as a rapidly developing, non-invasive neuroimaging technology [11,12], offers great potential for the detection of ASD in its early stages [13,14,15]. Considering that ASD affects normal neural connectivity between different brain regions of interest (ROIs), it is generally necessary to first estimate a functional brain network (FBN) based on fMRI and then treat it as a feature or biomarker to automatically distinguish subjects with ASD from normal controls (NCs) [16,17,18].

Over the past few decades, researchers have proposed a number of approaches to estimate FBNs from fMRI [19,20]. Several classic methods for FBN estimation include Pearson’s correlation (PC), sparse representation (SR) [21], mutual information (MI) [22,23], and correlation’s correlation (CC) [24]. In particular, these methods generally model the relationships between ROIs from different perspectives/views. For example, the PC method measures full correlation among ROIs, the SR strategy captures partial correlations, the MI method models nonlinear relationships, and the CC strategy encodes high-order relationships among ROIs.

Although they can estimate FBNs well in some scenarios, these methods only consider the association between ROIs from a single perspective, thus failing to capture the complex interactions among ROIs in the brain. To obtain effective representations of brain fMRI, several previous studies attempted to fuse multiview FBNs derived from rs-fMRI using different FBN estimation strategies [25,26,27,28,29,30]. For instance, Jie et al. [25] proposed thresholding of brain FBNs constructed using the PC method to generate multiple thresholded FBNs, followed by a multiple-kernel learning method to combine features of these FBNs for classification. Huang et al. [26] first constructed multiple FBNs for each subject with different levels of sparsity by setting different regularization parameters of an $l_{2,1}$-paradigm-based group-constrained sparse regression model. Then, they used a multi-kernel support vector machine (SVM) for classification based on selected features of each individual FBN. Gan et al. [27] proposed the construction of multiview FBNs by varying *k* values using the *k*-nearest neighbor (*k*-NN) algorithm, followed by an $L_{1}$-SVM for joint ROI selection and disease diagnosis based on the fused features of those multiview FBNs. Despite achieving good performance in fMRI-based brain disease diagnosis, these methods generally obtain multiview FBNs by setting different thresholds based on a single FBN estimation method, which essentially only model the same type of interconnections among ROIs at different levels.

In this paper, we propose a novel multiview FBN fusion method for fMRI-based ASD diagnosis, which can capture common and complementary fusion information from multiple views through a joint embedding strategy. Specifically, we first construct multiview FBNs based on several different strategies. We then model these FBNs through a third-order tensor in which each slice of the tensor represents the adjacency matrix (used to describe a certain relationship between ROIs) of an FBN from a single view. We further employ tensor decomposition [31] to learn the joint embedding of multiview FBNs in a latent space to capture view-shared and complementary features of fMRI data. Finally, we calculate the correlations between ROIs in the embedding space to obtain a new FBN for each subject, followed by a classification module for automated ASD diagnosis. Experimental results on rs-fMRI views of 184 subjects from the ABIDE dataset demonstrate the effectiveness of the proposed method in computer-aided ASD diagnosis.

Major contributions of this work are summarized as follows.

*First*, we propose fusion of multiview FBNs for fMRI-based ASD analysis based on different functional network estimation methods. Since these estimation methods naturally incorporate different prior knowledge among brain ROIs (such as partial correlation or sparsity prior), our method is expected to capture these rich and diverse relationship between ROIs, generating more reliable FBNs compared with conventional methods. *Second*, the proposed multiview FBN fusion is implemented in a low dimensional embedding space through tensor decomposition (rather than in the original high dimensional space of FBNs). This helps model the deal with potential common and complementary information among multiview FBNs and also eliminates some redundant information conveyed in different FBNs. *In addition*, the obtained common features of multiview FBNs naturally caters to an elegant theoretical explanation, i.e., the eigenvector centrality, which is a popular metric to evaluate the importance of nodes (i.e., ROIs) [32]. This will improve the interpretability of the proposed method in detecting disease-related functional connectivity abnormalities, thereby enhancing its utility in clinical practice.

The rest of this article is organized as follows. In Section 2, we briefly review four traditional methods for estimating FBNs. In Section 3, we introduce the materials and the proposed method. In Section 4, we describe the experimental setting and report experimental results. In Section 5, we discuss the influence of several key components of the proposed method and visualize discriminative functional connectivity features identified by our method in automated ASD diagnosis. Finally, we conclude this paper in Section 6.

## 2. Related Work

In this section, we briefly review several classical methods for functional brain network (FBN) estimation from four different perspectives, i.e., Pearson’s Correlation (PC), sparse representation (SR), mutual information (MI), and correlation’s correlation (CC).

### 2.1. Pearson’s Correlation

As reported in previous studies [33,34,35,36], Pearson’s correlation (PC) is one of the most widely used methods for FBN estimation based on fMRI data. It measures the *full correlation* between paired ROIs. For each subject, we suppose that the brain is parcellated into *n* ROIs according to a given atlas. The average blood-oxygenation-level-dependent (BOLD) signal of the *i*-th ROI is denoted as $x_{i}\in R^{v}$ (i=1,⋯,n), where *v* is the number of time points in the signal. The edge weight ($w_{ij}$) for the *i*-th and the *j*-th ROIs in a PC-based FBN can be calculated as follows:(1)$$\begin{array}{c} w_{ij}=\frac{{\left(x_{i}-{\bar{x}}_{i}\right)}^{T}\left(x_{j}-{\bar{x}}_{j}\right)}{\sqrt{{\left(x_{i}-{\bar{x}}_{i}\right)}^{T}\left(x_{i}-{\bar{x}}_{i}\right)}\sqrt{{\left(x_{j}-{\bar{x}}_{j}\right)}^{T}\left(x_{j}-{\bar{x}}_{j}\right)}} \end{array}$$
where ${\bar{x}}_{i}$ and ${\bar{x}}_{j}$ are the mean of $x_{i}$ and $x_{j}$, respectively. Generally, an FBN constructed by PC is a dense network [37], which may include noisy or useless edges. In practice, a threshold operation is usually employed to sparsify edges in the estimated FBN [25].

### 2.2. Sparse Representation

In contrast to the PC method, which measures the full correlation between two ROIs, sparse representation (SR) captures *partial correlation* by regressing out the confounding effect from other ROIs. Additionally, with the $l_{1}$ regularizer, the use of SR can naturally result in a sparse FBN. Specifically, the edge weight ($w_{ij}$ ) of the SR-based FBN can be calculated as follows:(2)$$\begin{array}{cc} \min_{w_{ij}}\; & \;\;\;\;\;\sum_{i=1}^{n}(∥x_{i}-\sum_{j\neq i}w_{ij}x_{j}{∥}^{2}{)+\lambda \sum_{j\neq i}|w_{ij}|}_{1}\\ s.t.\; & \;w_{ii}=0,\forall i=1,\cdots ,n \end{array}$$
where the first term in Equation (Equation 2) is a data-fitting term for modeling partial correlation between ROIs, the second term is an $l_{1}$ regularizer used to encode the sparsity prior of the FBN, and λ is the regularized parameter for controlling the balance between two terms in the objective function. The constraint $W_{ii}$ = 0 is used to avoid trivial solutions.

### 2.3. Mutual Information

The mutual information (MI) method measures shared information in time series data of two ROIs [22,23]; the edge weight ($W_{ij}$) in an MI-based FBN can be calculated as follows:(3)$$\begin{array}{c} w_{ij}=\sum_{i}\sum_{j}p\left(x_{i},x_{j}\right)\log\frac{p(x_{i},x_{j})}{p\left(x_{i}\right)p\left(x_{j}\right)} \end{array}$$
where $p(x_{i},x_{j})$ is the joint probability distribution of $x_{i}$ and $x_{j}$, and $p(x_{i})$ and $p(x_{j})$ are the marginal probability distributions of $x_{i}$ and $x_{j}$, respectively. In contrast to PC and SR, MI methods tend to capture *non-linear relationships* between ROIs. In particular, $W_{ij}$= 0 if $x_{i}$ and $x_{j}$ are independent or, equivalently, $p\left(x_{i},x_{j}\right)=p\left(x_{i}\right)p\left(x_{j}\right)$.

### 2.4. Correlation’s Correlation

The abovementioned methods, independent of linearity, calculate direct (or low-order) relationships between ROIs. Recent studies have shown that high-order interaction information between ROIs generally plays an auxiliary role in the early diagnosis of brain diseases [39,40]. Several methods for estimating high-order FBNs have been proposed in recent years [24,40,41], among which the correlation’s correlation (CC) method is widely used due to its simplicity [24]. The CC method generally involves two sequential steps. (1) PC is used to estimate a low-order FBN with the adjacency matrix $W={\left\{w_{ij}\right\}}_{i,j=1}^{n}$ for each subject. (2) Each column of the matrix (*W*) is treated as a new feature vector to calculate the high-order edge weights ($H_{ij}$), again using PC as shown in Equation (Equation 1). In this way, the CC method is expected to capture high-order relationships between paired ROIs.

## 3. Materials and Methods

In this section, we first describe the data used in our study and then propose a multiview FBN fusion method for automated ASD diagnosis, including its motivation, model formulation, and implementation details.

### 3.1. Data Preparation

In this study, we use rs-fMRI data from the largest site (i.e., NYU) of the ABIDE initiative [38], including 79 subjects with ASD and 105 normal controls (NCs). All the preprocessed fMRI data can be freely obtained on the ABIDE website (http://fcon_1000.projects.nitrc.org/indi/abide/ (accessed on 10 October 2022)). The demographic information of all subjects involved in this work is reported in Table 1.

The rs-fMRI data in the ABIDE dataset were obtained on a clinical routine 3.0 Tesla Allegra scanner using a standard echo-planar imaging sequence. The imaging parameters are shown as follows: TR/TE, 2000/15 ms; number of slices, 33; flip angle, $90^{\circ }$; voxel size, 3×3×4 mm${}^{3}$. To ensure signal stability, the first 10 volumes of each subject were removed from the rs-fMRI time course. The remaining volumes were then processed by the Data Processing Assistant for rs-fMRI-based (DPARSF) toolbox according to a recognized pipeline: (1) slice timing correction and head motion correction; (2) registration of the Montreal Neurological Institute (MNI) space with a resolution of 3×3×3 mm${}^{3}$; (3) regression of the nuisance signal, including ventricular, white matter, global signal, and motion parameters [42]; and (4) filtering with a 0.01–0.1 Hz band-pass filter to reduce the effects of heartbeat and respiration. Then, based on the automated anatomical labeling (AAL) atlas [43], the brain is parcellated into 116 regions of interest (ROIs), and the representative BOLD signal is extracted by the averaging strategy [44] from each ROI.

### 3.2. Proposed Method

#### 3.2.1. Motivation

Researchers have found that resting-state functional magnetic resonance imaging (rs-fMRI)-derived functional brain networks (FBNs) are a powerful tool for measuring and mapping brain activity [45]. Previous FBN estimation methods generally model connections between ROIs from a single perspective, thus encoding single prior information. The brain is a complex system, and FBNs estimated from a single perspective may have difficulty in capturing the subtle disruptions between ROIs caused by neurological disorders [26]. Therefore, we attempt to fuse multiview FBNs to provide a more comprehensive representation of the brain using shared and complementary information from different perspectives.

The most natural approaches for multiview network fusion are averaging, minimization, and maximization. Intuitively, these strategies should reduce the discriminative information of multiview FBNs. For example, averaging edge weights, especially those with positive and negative signs, easily leads to mutual cancellation of edge weights due to their opposite signs. Similarly, minimization and maximization are prone to result in more negative or positive connection weights, possibly leading to the loss of important edge weights, thus degrading the discriminative ability of fused FBNs. To verify this ability, we calculate the interclass distance of FBNs between the ASD group and the NC group, based on four different estimation methods (based on the data described in Section 3.1, the FBN matrices of all training subjects are obtained using different estimation methods, and the upper triangles of such matrices are pulled into vectors; then, all vectors are grouped into two categories (i.e., normal and patient) according to their known labels). Then, all vectors of each class are averaged separately as representative vectors of each class. Finally, we calculate the Euclidean distance between the representative vectors of the two classes as the final interclass distance. The results are shown in Table 2, based on which we can conclude that the fusion FBNs using the averaging, maximization, and minimization schemes obtain a relatively smaller interclass distance.

With the aim of improved fusion of multiview FBNs, we propose a novel joint embedding fusion scheme through tensor decomposition. In contrast to previous studies, our proposed scheme is implemented in a low-dimensional joint embedding space to construct the final FBN rather than in the original high-dimensional space of all FBNs. Such a joint embedding not only removes redundant information, including some noises or error correlations, but also captures the representative principle components of FBNs.

#### 3.2.2. General Framework

In Figure 1, we show the general framework of the proposed multiview FBN fusion method. *First*, we estimate the initial FBNs for each subject using four conventional methods, i.e., PC, SR, MI, and CC. Note that many other FBN estimation methods can also be used here, such as some improved strategies that incorporate specific prior information [36,46,47]. Since this paper is focused on fusing multiview FBNs, we choose four simple and representative estimation methods. *Then*, we stack the initially estimated multiview FBNs into a third-order tensor and apply the tensor decomposition method to jointly learn the common embedding of correlations in each FBN in the latent space. *Finally*, in the embedding space, we utilize PC (here, we use PC due to its simplicity and popularity. In principle, any existing method can be used to compute the correlations among the extracted principal components of the original FBNs to construct a new FBN, although this is outside the scope of the present study). Note that in the proposed framework, joint embedding plays a core role in capturing the common information of multiple FBNs. Therefore, we focus on the joint embedding step in the rest of this section.

#### 3.2.3. Proposed Joint Embedding

The proposed joint embedding strategy aims to map the node/ROI representations estimated by different methods into the same space to capture their potential common information. In particular, joint embedding is implemented as follows:(4)$$\begin{array}{cc} \underset{P,R}{\arg\min}\; & \;\frac{1}{2}\sum_{k=1}^{m}{∥A^{\left(k\right)}-PR^{\left(k\right)}P^{⊤}∥}_{F}^{2}+\frac{\alpha }{2}\sum_{k=1}^{m}{∥R^{\left(k\right)}∥}_{F}^{2}\\ \;s.t.\;\; & \;P^{⊤}P=I \end{array}$$
where $A^{\left(k\right)}$ is the $k_{th}$ slice of tensor A stacked by FBNs estimated by different methods. $PR^{\left(k\right)}P^{T}$ is an approximate decomposition of $A^{\left(k\right)}$ that is defined as follows: (5)$$\begin{array}{c} A^{\left(k\right)}\approx PR^{\left(k\right)}P^{⊤} \end{array}$$
where the matrix $P\in R^{n\times r}$ is the common factor shared among all $A^{\left(k\right)}\in R^{n\times n}$ that are FBNs, and $p_{i}$ (i.e., the $i_{th}$ row of *P*) represents the joint embedding of correlation of the *i*-th ROI in all FBNs, which seamlessly captures and integrates the inherent common information across all FBNs. Here, *n* is the number of ROIs, and *r* is the dimension of the embedding space. $R^{\left(k\right)}\in R^{r\times r}$ denotes the underlying interaction between ROIs of $A^{\left(k\right)}$. α is the regularization parameter, and ${∥.∥}_{F}$ is the Frobenius norm. In addition, the columns of *P* are orthogonal. The regularization term is introduced to avoid overfitting and to improve numerical stability [48,49].

In particular, the proposed model is highly *scalable* and can be effectively incorporated with any number of FBNs (i.e., $A^{\left(k\right)}$) constructed from different perspectives. This property makes our method feasible to capture comprehensive and complex relationships among brain ROIs. Interestingly, we find that this model has a clear theoretical explanation for jointly extracted common factors (*P*), as introduced below.

#### 3.2.4. Theoretical Explanation

In the proposed model, the matrix (*P*) is multiplied simultaneously on both sides of Equation (Equation 5) to obtain the following equation:(6)$$\begin{array}{c} A^{\left(k\right)}P\approx PR^{\left(k\right)} \end{array}$$

In graph theory, eigenvector centrality is a popular and important index to measure the importance of nodes [32]. Its formula is listed as follows:(7)$$\begin{array}{c} Ax=\lambda x \end{array}$$
(8)$$\begin{array}{c} EC\left(i\right)=x_{i} \end{array}$$
where *A* is the adjacency matrix of a graph, λ is the maximum eigenvalue of *A*, and *x* is the eigenvector corresponding to λ. The $i_{th}$ element (EC(i)) in the eigenvector *x* indicates the importance of the $i_{th}$ node of the graph.

According to the definition of eigenvector centrality, based on Equations (Equation 6) and (Equation 7), the proposed joint embedding is exactly the common maximum eigenvector of multiview FBNs, where each element indicates the importance of each ROI. Similarly, we further generalize the definition of eigenvector centrality. That is, when r>1, the commonality matrix (*P*) is composed of the top *r* eigenvectors of multiple FBNs, where each ROI has *r* features indicating its importance. Theoretical analysis further proves the feasibility of our proposed method.

#### 3.2.5. Optimization

Several different algorithms have been developed to date to solve Equation (Equation 4) [31,48,50]. In consideration of its generality and scalability, in this paper, we choose the RESCAL method [31] to decompose the tensor stacked by multiview FBNs and obtain the joint representation of multiview FBNs in the latent space. Specifically, *P* is initialized by the eigenvalue decomposition of $\sum_{k}\left(A^{\left(k\right)}+{A^{\left(k\right)}}^{⊤}\right)$, and $R^{\left(k\right)}$ can be initialized by any random matrix. The detailed optimization process is introduced as follows.

(a)**Update** P:

The update formula of *P* is obtained by RESCAL-alternating least squares (RESCAL-ALS) method [31], as shown below:(9)$$P\leftarrow \frac{\sum_{k=1}^{m}A^{\left(k\right)}P{R^{\left(k\right)}}^{⊤}+{A^{\left(k\right)}}^{⊤}PR^{\left(k\right)}}{\sum_{k=1}^{m}R^{\left(k\right)}{R^{\left(k\right)}}^{⊤}+{R^{\left(k\right)}}^{⊤}R^{\left(k\right)}+\alpha I}$$

(b)**Update**$R^{\left(k\right)}$:

The update formula of $R^{\left(k\right)}$ is as follows:(10)$$R^{\left(k\right)}=V\;\;\left(S\cdot U^{⊤}A^{\left(k\right)}U\right)V^{⊤}$$
where *U* and *V* are the left singular value matrix and the right singular value matrix of $P=U\Sigma V^{T}$, respectively. The sign “·” means dot product. In addition, *S* is defined as follows:(11)$$S=\left(\begin{array}{ccc} \frac{\Sigma_{11}\Sigma_{11}}{{\left(\Sigma_{11}\Sigma_{11}\right)}^{2}+\alpha } & \cdots & \frac{\Sigma_{rr}\Sigma_{11}}{{\left(\Sigma_{rr}\Sigma_{11}\right)}^{2}+\alpha }\\ \vdots & \ddots & \vdots \\ \frac{\Sigma_{11}\Sigma_{rr}}{{\left(\Sigma_{11}\Sigma_{rr}\right)}^{2}+\alpha } & \cdots & \frac{\sum_{rr}\Sigma_{rr}}{{\left(\Sigma_{rr}\Sigma_{rr}\right)}^{2}+\alpha } \end{array}\right)$$

To calculate the factor matrix, the Algorithm 1 performs alternate updates of *P* and all $R^{\left(k\right)}$ until $\frac{\sum_{k=1}^{m}{∥A^{\left(k\right)}-PR^{\left(k\right)}P^{⊤}∥}_{F}^{2}}{{∥\chi ∥}_{F}^{2}}$ converges to a small threshold (ϵ) or exceeds the maximum number of iterations. Here, χ is $\sum_{k=1}^{m}A^{\left(k\right)}$. After obtaining the common representation *P* of multiple FBNs in the latent space, we can obtain the reconstructed FBN through PC.

**Algorithm 1:** Algorithm of MJE**Input**: An× n × m: adjacency tensor; α: regularization parameter; *r*: rank of the latentrepresentation; tmax: the maximum number of iterations, ϵ THIS **Initialize**: *P* with the *r*largest eigenvectors of the Eigen decomposition of ∑k(A(k)+A(k)T); R(k) is initialized byany random matrices **While** not converged or t<tmax **do**
Update P according to Equation (11);Update R(k) according to Equation (14);t = t + 1;check the convergence conditions:∑k=1mA(k)−PR(k)PTF2χF2→ϵ or t>tmax**end**Reconstruct the FBN with PC for the potential representation of each ROI byW=PPT**Output**: Restructuring the FBN: *W*


In summary, our proposed algorithm has the following advantages. (1) Our algorithm fuses multiple perspectives of FBNs to provide a more comprehensive representation of the brain by exploiting shared and complementary information from different perspective. (2) Our algorithm is implemented in a low-dimensional embedding space. This helps to model the potential commonality and complementary information among the multiview FBNs and eliminate some redundant information passed between different FBNs. (3) Our proposed algorithm provides a reasonable explanation from the perspective of eigenvector centrality, thus enhancing its usefulness in clinical practice. (4) Although our work uses only four typical methods to build FBNs, our framework is scalable and can be extended to multiple FBNs.

The time complexity of updating the matrix (*P*) and the core tensor (R) is $O(m\left(r^{3}+nr^{2}\right)+pr)$. Here, p=nnz(A) refers to non-zero numbers in A. The time complexity of reconstructing the FBNs is $O(nr^{2})$. Therefore, the overall time complexity of our proposed approach is $O(M\left(r^{3}+nr^{2}\right)+pr+nr^{2})$.

#### 3.2.6. Classification

With the fused FBNs for each subject, we can perform ASD vs. NC classification. Specifically, we use edge weights of the FBN as features and a support vector machine (SVM) with linear kernels and a default parameter (*C* = 1) as a classifier.

## 4. Experiments

### 4.1. Experimental Setting

As mentioned in Section 2, we choose four representative methods, i.e., PC, SR, MI, and CC, to construct multiview FBNs. These FBNs are stacked into a tensor, and the tensor is then decomposed to obtain shared information for the fusion of multiview FBNs. For the PC, MI, and CC methods, their constructed FBNs are dense. Thus, we empirically select different thresholds to remove a proportion of edges in the range of [0%,10%,…,90%,99%] [46,47]. For SR, the sparsity of the estimated FBN can be controlled by the values of the regularization parameter that are searched in the range of $\left[2^{-5},2^{-4},\ldots ,2^{4},2^{5}\right]$ [47]. In the objective function of our method (see Equation (Equation 4)), the range of the regularization parameter α is [0.001,0.01,0.1,1,10,100,1000], and the reduced dimension of embedding space (*r*) is tuned in the set of [20,30,40,50,60,70,80,90,100,110].

Since only 184 subjects are involved in our experiments, we use leave-one-out (LOO) cross validation (CV) to obtain the final classification accuracy, as shown in Figure 2. In addition, we perform an inner loop of LOOCV to determine the optimal parameter values based on the training data. Specifically, we select the features of 182 subjects contained in a training set of 183 subjects to train the classifier and leave one subject to validate the performance of the trained classifier. The accuracy values of the 183 runs are recorded, and the parameter with the highest classification accuracy is selected as the parameter result of one cycle. Finally, the frequency of occurrence of each parameter is calculated, and the parameter with the highest frequency of occurrence is taken as the optimal parameter for the four different methods. The initial FBN corresponding to the optimal parameter is used as one of the FBNs to be fused at the end.

We report the classification performances of different methods on seven evaluation metrics, i.e., accuracy (ACC), specificity (SPE), sensitivity (SEN), positive predictive value (PPV), negative predictive value (NPV), class-balanced accuracy (BAC), and the area under the receiver operating characteristic curve (AUC), calculated as follows $ACC=\frac{TP+TN}{TP+FP+TN+FN}$, $SPE=\frac{TN}{TN+FP}$, $SEN=\frac{TP}{TP+FN}$, $BAC=\frac{SEN+SPE}{2}$, $PPV=\frac{TP}{TP+FP}$, and $NPV=\frac{TN}{FN+TN}$, respectively, where TP, TN, FP, and FN are the number of correctly predicted patients, correctly predicted normal controls, normal controls predicted as patients, and patients predicted as normal controls, respectively.

### 4.2. Comparison Methods

We compare the proposed method with several state-of-the-art fusion methods, including shallow fusion and deep fusion strategies. The comparison methods are as follows:•**MNER** [26]: This method uses the sparse regression model with group constraints to generate multiple sparse FBNs with different sparsity levels, followed by multiview FBN fusion via a multiview learning method.•**LORTA** [46]: This method assumes that FBNs have similar but not necessarily the same topology across subjects. It is implemented in a two-step learning framework. First, the FBNs are estimated according to conventional methods. Then, the estimated FCNs of all subjects are stacked into a tensor and refined by low-rank tensor approximation.•**BMGF** [27]: This method aims to fuse a fully connected FBN and a 1-nearest neighbor (1NN) FCN, taking into account the effects of intersubject variability and cross-subject heterogeneity.•**GraphCGC-Net** [51]: This method is a unified three-stage graph learning framework for brain disease diagnosis. First, it constructs a coarsened graph to obtain a critical graph structure using supervised multigraph clustering. A graph GAN is then used to generate the realistic brain networks based on the coarsened graph. It further finetunes the pretrained GCN by combining the generated and original graphs into a mixed training dataset.•**MVS-GCN** [30]: This method is a prior brain structure learning-guided multiview graph convolution network framework. It first constructs multiview coarsened brain network structures that are consistent for all the subjects and then implements multitask graph embedding learning to capture the intrinsic correlations among different views.•**MFC-PL** [52]: This method trains DNN models through unsupervised and supervised training steps to learn abstract feature representations of low-order and high-order FC patterns. Then, the learned multilevel abstract FC features are combined, and an ensemble classifier is trained on the fused features for brain disease classification.•**BrainGC-Net** [53]: This method improves the classification performance of the graph through three mechanisms. First, a priori subnetwork structure regularization is proposed to guide the pooling process and ensure accurate subnetwork identification. Then, a graph GAN model that focuses on both embedding and graph space is proposed based on the structure of α-GAN. In addition, a domain-consistent GCN model is proposed to alleviate the gap that exists between the real graph and the domain of the generated graph.

Among the abovementioned methods, MNER [26], LORTA [46], and BMGF [27] are shallow fusion methods, whereas GraphCGC-Net [51], MVS-GCN [30], MFC-PL [52], and BrainGC-Net [53] are deep fusion approaches.

In the experiments, for a fair comparison we adjust the above methods with parameters to obtain the best classification performance for them. The divisions of training and testing data are identical. In our method, we use edge weights of FBNs as the features for ASD vs. NC classification for each competing method. Because the adjacency matrix of the FBN is symmetric (because the SR model yields an asymmetric FBN, we symmetrize the FBN using the simple strategy of $W\leftarrow (W+W^{⊤})/2$), we only consider the upper triangular elements of the adjacency matrix, resulting in 6670 features for each participant. Furthermore, we employ a *t*-test to select more discriminative features by empirically setting the value of *p* to 0.01, 0.05, 0.001, and 0.005. Finally, an SVM with linear kernels and default parameters is used to perform the classification tasks.

### 4.3. Results

#### 4.3.1. Initial FBN Parameter Selection

In Figure 3, we report the optimal parameter choices for the four different methods based on a *t*-test with *p*-values of 0.01, 0.05, 0.001, and 0.005. Figure 3 shows that when the *p*-values are 0.01,0.05,0.001, and 0.005, the thresholds selected for PC are 40%, 80%, 80%, and 80%; the regularization parameters selected for SR are $2^{0}$, $2^{3}$, $2^{-4}$, and $2^{-1}$; the thresholds for MI are 90%, 70%, 50%, and 70%; and the thresholds for CC are 70%, 20%, 70%, and 80%, respectively.

#### 4.3.2. Results of ASD Identification

We report the results of all methods in the task of ASD vs. NC classification in Table 3. Based on the results presented in Table 3, we report the following observations. *First*, the proposed method outperforms the state-of-the-art methods in most cases. Specifically, in comparison with the three shallow fusion-based methods (i.e., MNER [26], LORTA [46], and BMGF [27]), the proposed method improves the performance by 3.81%, 5.98%, and 8.16%, respectively, in terms of ACC and by 8.40%, 7.43%, and 11.16%, respectively, in terms of AUC. This may benefit from the proposed joint embedding-based fusion, which captures more discriminative information of multiview FBNs from different views. Another possible reason is that our approach takes into account the complementary information of FBNs generated by different estimation methods. We can also see that our method compares to achieves better classification performance than the four deep fusion methods (i.e., GraphCGC-Net [51], MVS-GCN [30], MFC-PL [52], and BrainGC-Net [53]), with improvements of 2.72% and 6.53% in terms of ACC and AUC, respectively. This is possible because deep fusion methods tend to rely on a large amount of data [51], whereas this study includes a limited number of subjects. Overall, the results show that our multigraph fusion approach is feasible because it can exploit common and complementary information between multiple FBNs obtained by different estimation FBN methods.

## 5. Discussion

In this section, we investigate the influence of model parameters, multiview FBN fusion effectiveness, and the number of FBNs. Then, we visualize the most discriminative features identified by our method in automated ASD identification.

### 5.1. Sensitivity to Parameters

In this paper, we use the joint embedding strategy, which employs two parameters, i.e., the regularization parameter (α) and the embedding space dimension (*r*). We now discuss the effect of these two parameters on ASD vs. NC classification and report the experimental results of our method using different parameter values in Figure 4. In the experiments, we fix one parameter to observe the influence of another parameter on the classification results. Figure 4a shows the results of our method with different values of α when the embedding space dimension (*r*) is 50. It can be observed that our method achieves optimal classification performance when α=10. When the regularization parameter (α) is very large (e.g., α=100), the classification performance gradually decreases. The possible reason is that when α is too large, our model pays focuses excessively on the regularization terms and ignores the data-fitting term in the fusion model (see Equation (Equation 4)). In Figure 4b, we report the influence of the embedding space dimension (*r*) (when α=10). As shown in Figure 4b, our proposed method achieves good performance when *r* falls within the range of [40,50] but does not produce satisfactory results when r=110. The reason may be that when the embedding space dimension (*r*) is too large, some noise may be included, negatively affecting classification performance.

### 5.2. Influence of Proposed Fusion Strategy

In this section, we compare the proposed multiview FBN fusion approach with its degenerated single-view variants (i.e., PC, SR, MI, and CC) based on LOOCV. The comparison results are reported in Table 4. As shown in Table 4, our proposed multiview FBN fusion method is superior to the other four methods in terms of performance in most cases. For example, with *p*-values of 0.001, and 0.005, our method achieves an accuracy of 74.46%, 8.16%, and 5.44% higher than the second-best method (i.e., CC and PC), respectively. These results further confirm that considering the rich and complementary information in multiview FBNs (as we do in this work) helps boost fMRI-based classification performance.

### 5.3. Influence of Number of FBNs

Based on the averaging fusion and our proposed method, we evaluate the influence of the number of FBNs on the classification performance with a *p*-value of 0.001 in Table 5. With different numbers of FBNs to be fused, we iterate different FBN combinations. Then, the classification accuracy of the different combinations with the same number of FBNs are averaged as the final results. Table 5 shows that our method obtains consistently better performance than the averaging fusion method, with a steady improvement as the number of FBNs increases. This may be because the proposed joint embedding fusion method can capture more complementary information than the averaging strategy.

### 5.4. Identified Discriminative Features

As mentioned in the previous section on the experimental setting, we use functional connections between ROIs as fMRI features to distinguish subjects with ASD from NCs. To visualize the disease-associated features, we select the most discriminative (i.e., top 52) features based on a *t*-test with a *p*-value of 0.001, as shown in Figure 5. The width of each arc represents the discriminative power of the corresponding functional connection. The colors of the arcs are generated randomly to provide visual clarity. As shown in Figure 5, the ROIs associated with the most discriminative features include the right–middle frontal gyrus, the right hippocampus, and the right amygdala. Some of these ROIs have been widely reported in previous studies on ASD diagnosis [54,55,56]. The result further confirms that our approach is reliable to discover fMRI biomarkers for automated ASD identification.

## 6. Conclusions

In this paper, we present a multiview FBNs fusion strategy through joint embedding for fMRI-based ASD identification. In contrast to traditional fusion strategies, the proposed method first stacks all differently estimated FNBs into a tensor and then performs tensor factorization to learn joint embedding to capture the potential common information across multiview FBNs. In addition, the obtained common representation naturally caters to an elegant theoretical explanation of eigenvector centrality. Experimental results obtained on the rs-fMRI data of 184 subjects show that the proposed approach achieves competitive performance relative to several existing methods for ASD disease diagnosis. In the future, we will generalize the proposed scheme to deep learning models, as these methods only focus on capturing the multilinear common relationships among multiview FBNs and ignore their potential nonlinear relationships.

## Figures and Tables

**Figure 1 jpm-13-00251-f001:**
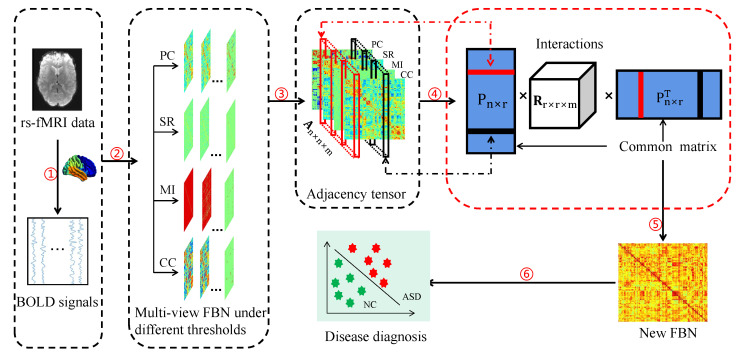
Illustration of the proposed multiview functional brain network fusion method, including six major parts: (1) rs-fMRI preprocessing; (2) estimation of initial functional brain networks (FBNs) based on four strategies, i.e., Pearson’s correlation (PC), sparse representation (SR), mutual information (MI), and correlation’s correlation (CC); (3) selection of the initial FBNs under the optimal parameter; (4) factorization of the tensor stacked by the selected FBNs to obtain the common matrix (*P*) of different FBNs; (5) construction of a new FBN based on *P* for each subject; and (6) disease diagnosis.

**Figure 2 jpm-13-00251-f002:**
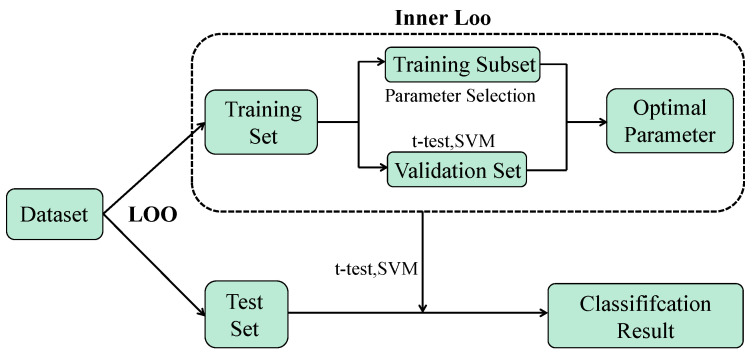
The cross-validation mechanism used in our experiments includes the internal LOO method to determine the best parameters and the external LOO method to obtain classification results.

**Figure 3 jpm-13-00251-f003:**
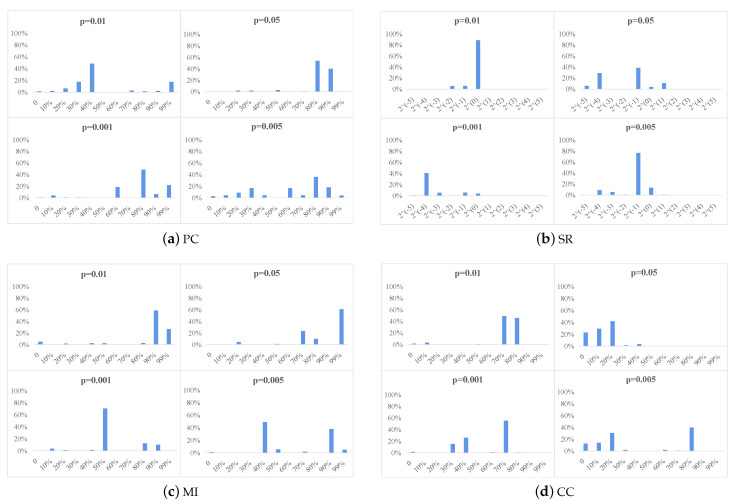
Frequencies of the optimal *p* values selected in an inner loop for the four different methods based on a *t*-test with p={0.01,0.05,0.001,0.005}. The horizontal axis indicates the multiple thresholds for the different methods, and the vertical coordinates indicate the frequencies of occurrence of the different thresholds.

**Figure 4 jpm-13-00251-f004:**
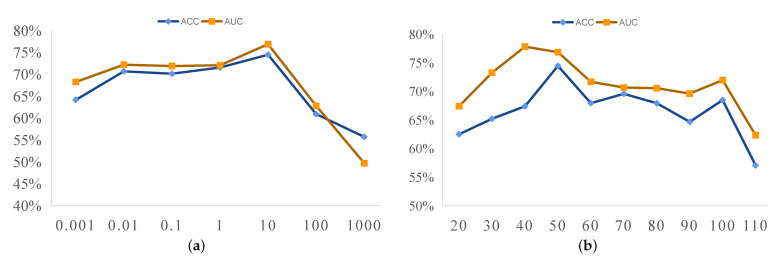
Classification results (ACC and AUC) of our proposed method according to different parameters based on a *t*-test with a *p*-value of 0.001. (**a**) Influence of the regularization parameter (α) on the model classification results; (**b**) effect of the embedding space dimension (*r*) on the model classification results.

**Figure 5 jpm-13-00251-f005:**
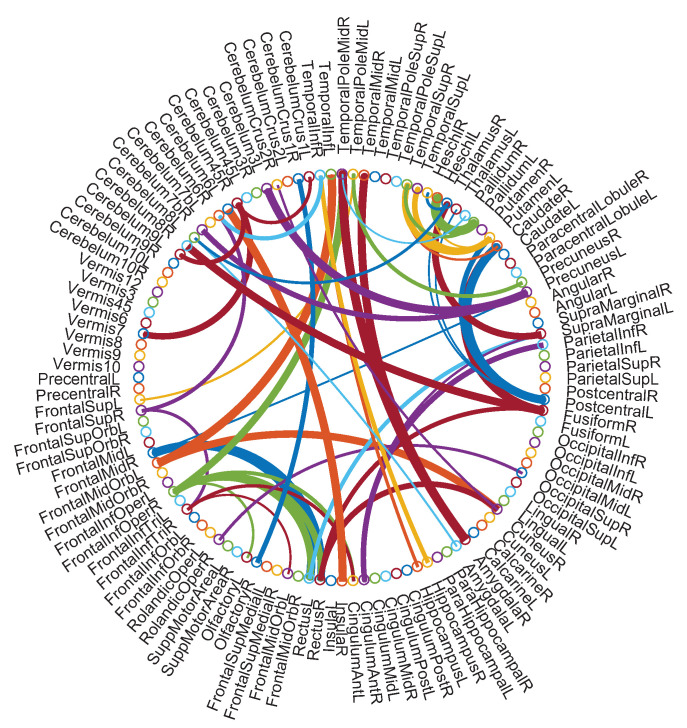
The most discriminative features detected by the proposed method in ASD vs. NC classification based on an AAL template. This figure was created using circularGraphtool (http://www.mathworks.com/matlabcentral/fileexchange/48576-circulargraph (accessed on 10 October 2022)).

**Table 1 jpm-13-00251-t001:** Demographic and clinical information of subjects at the NYU site from the ABIDE dataset [38]. Values are reported as mean ± standard deviation. M/F: male/female; MMSE: Mini-Mental Examination; GCDR: Global Clinical Dementia Rating; FIQ: Full-Scale Intelligence Quotient; VIQ: Verbal Intelligence Quotient; PIQ: Performance Intelligence Quotient.

Dataset	Class	Gender (M/F)	Age (Years)	FIQ	VIQ	PIQ
ABIDE	ASD	68/11	18.58±11.45	107.92±3.15	105.81±1.23	108.81±2.10
	NC	79/26	19.13±11.85	113.15±2.45	113.13±1.15	115.07±2.08

**Table 2 jpm-13-00251-t002:** Interclass FBN distance between ASD patients and normal controls based on different methods. Aver: averaging fusion; Min: minimization fusion; Max: maximization fusion.

Method	PC	SR	MI	CC	Aver	Min	Max	Ours
Interclass FBN Distance	9.20	2.52	6.41	12.26	5.88	6.87	6.02	13.24

**Table 3 jpm-13-00251-t003:** Classification results (mean ± standard deviation) of six methods in the task of ASD vs. NC classification, with best results shown in bold.

Method	ACC (%)	SEN (%)	SPE (%)	BAC (%)	PPV (%)	NPV (%)	AUC (%)
MNER [26]	70.65	58.82	74.29	66.66	61.98	74.29	73.32
LORTA [46]	68.48	**71.70**	64.10	67.90	64.93	74.29	74.29
BMGF [27]	66.30	60.76	70.48	65.62	60.76	70.48	70.56
GraphCGC-Net [51]	71.74	63.83	78.26	71.05	**75.00**	**78.26**	77.42
MVS-GCN [30]	67.93	58.23	75.24	66.73	63.89	70.54	71.14
MFC-PL [52]	66.74	56.54	74.95	65.75	63.10	63.10	69.70
BrainGC-Net [53]	77.43	59.30	51.20	58.52	59.17	64.30	74.83
Ours	**74.46**	64.56	**81.90**	**73.23**	72.86	75.44	**81.72**

**Table 4 jpm-13-00251-t004:** Classification results (mean ± standard deviation) of the proposed method and four single-view methods (i.e., PC, SR, MI, and CC) based on different *p*-values involved in the *t*-test. CV: cross validation.

CV	*p*-Value	Method	ACC (%)	SEN (%)	SPE (%)	BAC (%)	PPV (%)	NPV (%)	AUC (%)
LOOCV	*p* = 0.01	PC	66.85	65.82	67.62	66.72	60.47	72.45	74.56
		SR	66.31	49.37	79.05	64.21	63.93	67.48	71.78
		MI	57.07	36.71	72.38	54.54	50.00	60.32	57.18
		CC	65.76	58.23	71.43	64.83	60.53	69.44	72.68
		Ours	**73.91**	**65.82**	**80.00**	**72.91**	**71.23**	**75.68**	**75.68**
	*p* = 0.05	PC	**67.39**	59.49	73.33	**66.41**	**62.67**	**70.64**	**71.79**
		SR	59.78	46.84	69.52	58.18	53.62	63.48	58.64
		MI	55.43	41.77	65.71	53.74	47.83	60.00	62.69
		CC	66.85	58.23	**73.33**	65.78	62.16	70.00	69.99
		Ours	66.30	**60.76**	70.48	65.62	60.76	70.48	70.56
	*p* = 0.001	PC	66.30	**65.82**	66.67	66.24	59.77	72.16	70.17
		SR	63.04	51.90	71.43	61.66	57.75	66.37	64.48
		MI	64.13	62.03	65.71	63.87	57.65	69.70	72.56
		CC	70.11	64.56	74.29	69.42	65.38	73.58	78.52
		Ours	**74.46**	64.56	**81.90**	**73.23**	**72.86**	**75.44**	**81.72**
	*p* = 0.005	PC	69.02	68.35	69.52	68.94	62.79	74.49	73.25
		SR	67.39	48.10	**81.90**	65.00	66.67	67.72	70.61
		MI	55.43	41.77	65.71	53.74	47.83	60.00	62.69
		CC	69.57	63.29	74.29	68.79	64.94	72.90	77.37
		Ours	**74.46**	**68.35**	79.05	**73.70**	**71.05**	**76.85**	**76.85**

**Table 5 jpm-13-00251-t005:** Influence of the number of FBNs based on a t-test with a *p*-value of 0.001.

Method	Number of FBNs		
	Two-View FBNs	Three-View FBNs	Four-View FBNs
Averaging Fusion	62.74%	63.99%	64.77%
Ours	71.73%	72.14%	74.41%

## Data Availability

The dataset used in this article can be freely obtained from the ABIDE website (http://fcon1000.projects.nitrc.org/indi/abide/ (accessed on 10 October 2022)).

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
