# Peer review of "Fusing Multiview Functional Brain Networks by Joint Embedding for Brain Disease Identification"

_jpm, 2023, doi:10.3390/jpm13020251_

Round 1
Reviewer 1 Report
The paper addresses an important problem and achieves meaningful results. However, the main weakness of the paper lies in its lack of originality and novelty. The following suggestions may be considered to enhance the quality and clarity of the manuscript.
1- Abstract is comprehensive and well written, but it needs improvements e.g. case study can be described a little more at the end of the paragraph.
2- The motivation is not clear. Why did this work? Is any problem does it address that the previous methods cannot?
3- This area is rapidly evolving, and new papers have been published. Therefore, some state-of-the-art (i.e., of 2022) papers should be taken into account:
https://www.mdpi.com/1424-8220/21/6/2222
https://www.sciencedirect.com/science/article/abs/pii/S0306987720308689
https://www.sciencedirect.com/science/article/pii/S0020025522007332
4- In the related works, existing studies can also be summarized in a tabular form to improve readability
5- Novelty of the algorithm needs to be incorporated
6- The authors provided all experimental details for reproducing the results. But it seems that the comparison of experiments is not satisfactory. Please compare your results with some state-of-the-art studies in a separate table.
7- Language also needs improvement
Reviewer 2 Report
The work reported by Wang et al represents an important addition to the literature that presents a multi-view FBNs fusion strategy. I recommend publishing after addressing the points below:
1. The introduction lacked citations of adequate references to give readers sufficient background on the topic.
2. The quality of Fig. 1 needs to be improved.
3. Text size needs to be increased in Fig. 2 and 3
4. Can the authors include error bars to the data presented in Fig. 4?
Reviewer 3 Report
Dear Authors
Congratulations on your work. Your manuscript is very well written and easy to follow and understand. I found it a very interesting approach.
Best regards,
Author Response
Thank you for this valuable comment.